🔓 | **Open Peer Review** | Food Microbiology | Research Article

# Microbiome characterization of two fresh pork cuts during production in a pork fabrication facility

A. E. Asmus,[1,2] T. N. Gaire,[1] K. J. Schweisthal,[2] S. M. Staben,[2] N. R. Noyes[1]

**ABSTRACT** The goal of this study was to characterize the microbial profile of two different fresh pork cuts, bootjack (BJ) trim and tenderloin (TL), through a 16S rRNA sequencing workflow developed specifically for investigating low-biomass fresh meat within a commercial production schedule. Additionally, this study aimed to determine a baseline *Salmonella* prevalence and enumeration profile across these two fresh pork cuts. Results showed that microbiome diversity was different between the BJ and TL, and also differed significantly by processing date. The relative abundance of key bacterial genera associated with food safety and spoilage was also different between the two meat types. However, over the course of the production shift, changes in the meat microbiome were limited in both the BJ and TL. The crude prevalence and enumerated burden of *Salmonella* were lower than what has been previously reported in similar fresh pork cuts, and all of the *Salmonella*-positive samples occurred on just two processing windows of 1–2 days each. Taken together, the results of this study suggest that the microbial profile of two fresh pork cuts is significantly different even within the same plant at the same time points, and that day-to-day variability within the production process likely influences both the fresh pork microbiome and *Salmonella* profile of these two meat types.

**IMPORTANCE** Modern pork processing involves a series of processes that begin with the handling and transport of the live animals, proceed through harvest and fabrication, and end with the packaging and distribution of fresh pork to the consumer. Each step in this process can alter the microbial community of fresh pork and influence the meat's safety and shelf life. However, little is known about the microbial ecology of individual, unprocessed pork cuts and if the diversity of the meat microbiome remains consistent throughout a production schedule. Additionally, the crude prevalence and enumeration of *Salmonella* have not been well established for individual fresh pork cuts throughout a production schedule. A more thorough understanding of the microbial profile at different stages of pork production will help processors determine processing steps that impact the microbial characteristics of fresh pork. This insight will help processors implement targeted intervention strategies to enhance food safety and quality.

**KEYWORDS** fresh pork, microbiome, *Salmonella*, 16S rRNA

A commercial pork processing plant is a complex network of equipment, automation, human activity, and processes, all of which work together to efficiently fabricate a pork carcass into multiple primal, subprimal, and trim cuts. Each step in the fabrication process, and the environmental inputs that exist at each step, can influence the microbial ecology and *Salmonella* status of fresh meat (1–5).

Investigating these influences is challenging, particularly in the context of an operating processing plant, in which the environment and incoming carcasses change rapidly and continuously, without pause. The opening of a new pork processing plant

**Peer Reviewer** Anuradha J. Punchihewage Don, University of Maryland Eastern Shore, Princess Anne, Maryland, USA

Address correspondence to N. R. Noyes, nnoyes@umn.edu.

This work was funded by Hormel Foods Corporation. A.E.A., S.M.S., and K.J.S. are employees of Hormel Foods Corporation.

presents a unique opportunity to observe the establishment of the plant microbiome, which was found to be an incredibly dynamic process. Specifically, microbiome changes were detected across multiple locations in the plant as it transitioned from construction, to the first 50 days of operation, and finally to +500 days of operation (6). These results suggest that the environmental microbiome of a pork plant is temporally dynamic over long time periods that span pre- and post-construction and operation of a new facility. Across much shorter timescales, the relative abundance of microbes and microbial genes associated with stress resistance also fluctuates, for example at different steps within the pork harvest process (7, 8). In addition to temporal changes, the microbial ecology of environmental samples has been reported to differ by meat cut and fabrication pathway, suggesting an influence of meat cut on the microbiota of processing surfaces (4). However, little has been reported regarding the temporal dynamics of fresh pork meat microbiomes across a daily production schedule and across different fabrication processes.

The overall prevalence of *Salmonella* in fresh pork cuts has been established through a nationwide baseline study (9) in which multiple processing plants and finished products were categorized by overall processing plant capacity (i.e., annual pounds of pork produced) and by finished product category (i.e., intact, non-intact, comminuted). While this study was able to establish a baseline *Salmonella* prevalence for each of these categories, it did not account for potential daily variation that may be observed at individual processing plants and at the individual raw material level. A recent study conducted by Bueno López et al. (1) showed that the prevalence and enumeration of *Salmonella* were variable at different points in the harvest process and between meat cut, providing evidence that the presence of *Salmonella* in a pork processing facility is dynamic across processing steps and individual meat cuts.

The current research into fresh pork microbial ecology and *Salmonella* has typically been limited to either (i) environmental sampling of the processing line, (ii) select packaged meat cuts exposed to different environments or formulations over the course of shelf life, or (iii) sampling of the processing line on multiple days but not consecutively and not at the same point within the processing shift on each sampling day. Therefore, there are critical gaps in our current knowledge of the microbial ecology, *Salmonella* prevalence and enumeration of fresh pork cuts, particularly in regard to short-term temporality and differences between fabrication lines within a single plant in the same time frame. The objective of this pilot study was to fill these gaps by characterizing the microbial ecology, *Salmonella* prevalence and enumeration of two different meat types processed on two different fabrication lines, throughout a processing shift, and across multiple consecutive and non-consecutive production dates, all within the same processing plant. To enable this overall objective, this study also developed methods and procedures that allow for multiple microbiological and molecular assays to be performed on a single meat sample, within the constraints of commercial production schedules.

## MATERIALS AND METHODS

### Study design and sample collection

This observational study utilized two individual fresh pork raw materials sampled on a routine basis from a large-scale pork processing facility that harvested approximately 10,000 market hogs per shift. The plant incorporated a 25 ppm chlorine carcass rinse after evisceration and prior to entering the carcass chiller. No other antimicrobials were used on the carcass or fabrication lines during normal production. Fresh pork meat types analyzed included the whole muscle tenderloin (TL) and Bootjack Trim (BJ). These two raw materials were handled on two separate fabrication lines in the pork processing facility. The TL was a component of the whole bone-in loin primal cut that was initially binned into combo bins on a separate fabrication line and then sequentially brought to the loin boning line for further fabrication. The TL was removed as one of the first steps in the fabrication of the whole boneless loin into the loin subprimal cuts. The BJ was

part of the belly primal cut and was removed via a perpendicular cut lengthwise from the posterior end of the belly as it was being right sized for further processing. Due to timing differences in fabrication lines and processes, it is highly unlikely that the BJ and TL samples collected on a given sampling date originated from the same pig or farm. Sampling of both fresh pork raw materials occurred 4 days per week (Monday, Tuesday, Wednesday, and Thursday) for two consecutive weeks. This 2-week sampling designated one testing period, followed by 1 week of no sampling prior to the start of the next period. Sampling periods spanned the months of June, July, and August of 2022.

One individual meat sample of both the BJ ($N = 71$) and TL ($N = 72$) was removed from each fabrication line during normal production at the beginning, middle, and end of the first shift of each day. Sampling at the beginning of the first shift occurred at approximately 6:00 a.m., middle at approximately 10:00 a.m., and end at 1:30 p.m. Each TL sample consisted of one whole, boneless, untrimmed TL (approximately 900 g) that was aseptically removed directly from the fabrication line immediately after it was excised from the whole boneless loin and placed in a sterile 7.5" × 15" whirl-pak bag (Whirl-Pak, Cat. No. B01451). Each BJ trim sample consisted of approximately 600 g of BJ trim that was aseptically removed from a transfer conveyor that was used to transport the BJ trim to a combo bin after it had been separated from the belly and placed in a sterile 7.5" × 15" whirl-pak bag (Whirl-Pak, Cat. No. B01451). All samples were placed in a cooler with ice packs and immediately transferred to the Hormel Foods Research and Development Laboratory for refrigerated storage (4.5°C) overnight. Total transit time between the plant and the laboratory was approximately 15 minutes. Sample preparation for microbiology analysis of both the BJ and TL samples occurred on the day following sample collection.

## Microbiological analysis–aerobic plate count

From each collected sample, an 11 ± 0.1 g piece of meat was aseptically removed and placed in a sterile whirl-pak filter bag (Whirl-Pak, Cat. No. B01348). Sterile Butterfield's Phosphate Buffer (99 mL) (World BioProducts, Cat. No. NLD-99BFD) was added to the meat and stomached on high speed for 30 seconds (AES Laboratore, Smasher, Cat. No. AESAP1064). Serial dilutions were made using 1 mL sample into 9 mL Butterfield's Phosphate Buffer tubes (World BioProducts, Cat. No. FRV-9BFD). Approximately 15–20 mL of Plate Count Agar (Neogen Culture Media, Cat. No. NCM0010B) was poured onto each sterile plate, along with the 1 mL of (diluted) sample. After allowing the plates to solidify at room temperature for approximately 1 hour, the plates were then incubated at 35°C ± 2°C for 40–48 hours. Countable range was considered 30–300 colonies per plate. Biological significance was determined to be ±0.5 CFU/g, based on historical data from internal, non-published shelf life studies.

## Microbiological analysis–*Salmonella* detection and enumeration

*Salmonella* analysis was conducted using the Hygiena-BAX System Real Time *Salmonella* PCR Assay and Hygiena SalQuant v 3.6 platform (Cat. No. KIT2006) following the AOAC Certification No. 081201 protocol. A 375 ± 2 g meat sample was aseptically removed and placed into a 15" × 15" sterile whirl-pak filter bag (Whirl-Pak, Cat. No. B01525). A 1,500 mL of 42°C prewarmed MP Media (BAX Systems, Cat. No. MED2003) was aseptically added to the filter bag and sample. Each bag was vigorously shaken by hand for 1 minute by shaking the bag back and forth. Samples were then incubated at 42°C for primary enrichment for *Salmonella*. To generate a sample for potential enumeration with a limit of detection (LOD) of 1 CFU/g, a 25 mL aliquot was collected from the primary enrichment at 6 hours ± 20 minutes of incubation, placed into a 50 mL conical tube (MTC Bio, Cat. No. C2603), and held at 4.5°C overnight. The remaining sample was incubated at 42°C for an additional 16–18 hours (22–24 hours total) for prevalence testing. A negative and positive pathogen control (*Listeria monocytogenes:* ATCC 15313, *Salmonella* Abaetetuba: ATCC 35640, *Escherichia coli* O157:H7: ATCC 43888) was also incubated alongside the meat samples at 42°C for 22–24 hours on each testing day, and

analyzed on each BAX PCR run. BAX Lysis reagent was premade in cluster tubes and held at 4.5°C, with a shelf life for the lysis reagent of 2 weeks.

After incubation, 5 µL of each sample was added to the lysis reagent, and tubes were placed in a programmed heating block to 37°C for 20 minutes, then 95°C for 10 minutes, followed by cooling to 4.5°C. Immediately following lysis, 30 µL of the lysate was added to a single well in a 96-well plate, containing one PCR tablet per well. PCR was conducted using the BAX Q7 thermocycler. Based on the cycle threshold, an estimated *Salmonella* enumerated CFU/g (LOD 1 CFU/g) was determined if the sample was PCR positive based on the 6-hour incubation. If the sample was negative based on the 6-hour-enriched aliquot but positive based on the 24-hour-enriched sample, the estimated *Salmonella* enumeration is <1 CFU/g.

*Salmonella* prevalence was calculated as the number of positive meat samples out of the total number of meat samples. Comparisons between meat types and shift times for *Salmonella* prevalence were made using the Pearson chi-squared test in R (*chisq.test* function, *stats* v.4.3.2 package). Prevalence results were displayed using the *gt* v0.10.0 package (10). Visualization of *Salmonella* enumeration and *Salmonella* positives by processing dates was generated using *ggplot2* v.3.5.0 package (11).

## DNA sample preparation, DNA extraction, library preparation, and sequencing

Meat rinsates from both the BJ and TL were used for DNA extraction. Immediately after the meat samples were aggressively hand massaged as part of the BAX System SalQuant procedure (see above), a 38 mL aliquot of the rinsate was portioned into a 50 mL conical tube and immediately centrifuged at 13,000 × *g* for 12 minutes at 4°C. The supernatant was removed, and the remaining pellet was retained and stored at −80°C until DNA extraction. Approximately 20 minutes total elapsed between the addition of the MP media to the meat and the storage of the rinsate pellet at −80°C.

All meat rinsate pellet samples were subjected to DNA extraction using the DNeasy PowerFood Microbial Kit (Qiagen, Cat. No. 21000-100, 0.5 mL bead tube, Hilden, Germany) and automated on the QiaCube Connect instrument (Qiagen, Cat. No. 9002864, Hilden Germany). Briefly, meat rinsate pellets were thawed for 20 minutes in a biosafety cabinet previously sterilized with UV light and 70% ethyl alcohol. Once thawed, 450 µL prewarmed (55°C) MBL reagent was added to each rinsate pellet and vortexed at high speed until the pellet was homogeneously suspended. The remaining DNA extraction procedures continued according to the manufacturer's instructions. The QiaCube instrument was run 13 times, with a maximum sample throughput of 12 samples per run. One negative control was included at the beginning of each extraction day (4 days total) and consisted of 450 µL of prewarmed MBL reagent with no sample that proceeded through the rest of the PowerFood extraction procedure alongside the meat samples. A single positive control that contained a known composition of eight bacterial and two fungal species (Zymo Research Corp., D6310) was included as an internal control on the last extraction run. All sample extractions were randomly ordered, and 20 µL of each extracted sample was transferred onto PCR plates (two total). Both plates were covered with an adhesive seal and submitted to the University of Minnesota Genomics Core (UMGC) for library preparation and sequencing.

Extracted DNA of 12 random samples was quantified using fluorometry (Invitrogen Qubit 4 Fluorometer, Cat. No. Q33238). The 16S rRNA gene copy number in each sample was determined using qPCR with both 25 and 30 PCR cycles to ensure adequate copy number of the 16S rRNA gene present for sequencing, with 30 cycles ultimately being used for amplification. Libraries were prepared by amplifying the V3-V4 region of the 16S rRNA gene, using primer V3_375F_Nextera: CCTACGGGAGGCAGCAG and V4_806R_Nextera: GGACTACHVGGGTWTCTAAT (12). Sterile, molecular-grade water negative controls for library preparation (*N* = 2) were also included during the library preparation process. Prepared libraries were sequenced to an expected sequencing depth of 100,000 paired

reads per sample on an Illumina MiSeq instrument (Illumina Inc, San Diego, CA, USA) using a 600 (2 × 300 base pair) cycle reagent kit (Illumina Inc, San Diego, CA, USA).

## Bioinformatics

Bioinformatic analyses were conducted using R Statistical Software (v. 4.3.2; R Core Team 2021), and plots were generated using the *ggplot2* package. Adapter sequences and primers were first removed from the raw sequences using *Cutadapt* v.4.0 (13). Raw sequences were then processed through the *DADA2* (v1.30.0) pipeline for quality filtering, denoising, and microbial inference (14). Forward and reverse reads were truncated to a length of 240 and 230 base pairs, respectively, based on the distribution of quality scores. Forward and reverse reads with expected error rates exceeding 3 and 4 base pairs, respectively, were discarded. Forward and reverse reads were merged, and chimeras were identified and removed using the *removeBimeraDenova* function. The merged, chimera-free amplicon sequence variants (ASVs) were aligned to the SILVA reference database v 138.1 (15) for taxonomic assignment. The ASV count matrix, taxonomy table, and sample metadata were combined to create a phyloseq object for further microbiome analysis using *phyloseq* (v.1.46.0) R package (16).

The positive control sequences were aligned to the manufacturer's reference database (https://s3.amazonaws.com/zymo-files/BioPool/ZymoBIO-MICS.STD.refseq.v2.zip). Raw sequences from the positive control (i.e., mock) were processed through *Cutadapt* and *DADA2* as described above. Analysis of the sequences occurred after QC and Filtering in *DADA2*, and they were not subjected to *decontam* for the removal of potential contaminants. The theoretical composition of the mock sample was compared visually to the observed composition of the positive control, by relative abundance of each feature present in the positive control.

## Identification and removal of contaminants

Identification and removal of potential sequence contaminants from DNA extraction and library preparation were conducted using the *decontam* v1.22.0 (17) package in R. The phyloseq object was subjected to three methods of contaminant identification: frequency, prevalence, and combined. A threshold = 0.1 was used in the *isContaminant* function for the frequency method, a threshold = 0.5 for the prevalence method, and a threshold = 0.4 for the combined method. The number of contaminant ASVs identified, overall prevalence of these ASVs in the samples, and relative abundance of the individual ASV contaminants identified in each control and sample were used to select the most appropriate contaminant identification method. ASVs identified as contaminants were removed from the final phyloseq object. Visualization of contaminants for the frequency, prevalence, and combined methods was conducted utilizing the methods and base R code provided by Dean et al. (18).

## Microbiome analysis

The contaminant-free phyloseq object was subsetted to only include ASVs that were identified as either Bacteria or Archaea at the kingdom level. Alpha diversity was estimated by calculating observed richness and Shannon diversity index using the *estimate_richness* function in the *phyloseq* R package. Pielou's evenness was calculated using the following equation: Evenness = {Shannon/log(Observed)}. Differences in alpha diversity and aerobic plate count by meat type, production schedule (i.e., production date, day of the week, shift time), and sequencing factors (fixed effects) were assessed using a linear regression model (*lm* function, *stats* v.4.3.2 package) with the final model chosen using a stepwise algorithm based on Akaike information criterion (AIC) from the *step* function in the *stats* package in R. A generalized linear model (*glm*) based on the best-fit linear model was used to test the interaction of meat type (BJ vs TL) and production schedule (shift time, date, week, and period) and sequencing factors (extraction batch, 16S qPCR copy number, sequencing plate) as fixed variables to the

dependent variables (i.e., APC, alpha diversity metrics) using the Type III *Anova* function in the *car* v3.1.0 package in R. For variables that significantly improved model fit, estimated marginal means were calculated, and pairwise comparisons were made using the *emmeans* function from the *emmeans* v.1.10.0 package in R. Plots to visualize estimated marginal means were generated using the *ggplot2* package.

For beta diversity, ASV counts were first normalized using cumulative sum scaling (CSS) from the *cumNorm* function in the *metagenomeSeq* v. 1.43.0 package. Bray-Curtis dissimilarity distances were calculated using the *distance* function in the *phylsoeq* package. Non-metric multidimensional scaling (NMDS) was used to ordinate the Bray-Curtis dissimilarity distances using the *ordinate* function in the *phyloseq* package. The effect of meat type and production schedule was tested using permutational multivariate analysis (PERMANOVA) using the *adonis2* function in the *vegan* v2.6.4 package. Both individual and interaction factors were included in the PERMANOVA model for meat type and shift time. Shift time was subsetted by individual meat type prior to PERMANOVA analysis. NMDS and spider plots were generated using the *ggplot2* package. Associated $R^2$ values for key variables were reported in a table using the *gt* package.

Abundance was calculated as the number of sequencing reads associated with each ASV within each sample. Each ASV was grouped at both the phylum and genus levels using the *tax_glom* function in the *phylsoeq* package. For relative abundance, ASV counts were transformed using the *transform_sample_counts* function (function(×) ×/sum(×)*100) in the *phyloseq* package. Abundance plots were generated using the *ggplot2* package.

Differential abundance of ASVs present was calculated using the *MaAsLin2* v1.16.0 package (19) by specifying a multivariate model with meat type and shift time as fixed-effect variables. Briefly, ASV counts were first normalized using CSS from the *cumNorm* function in the *metagenomeSeq* v. 1.43.0 package and grouped at the genus level using the *tax_glom* function in the *phyloseq* package. Data were analyzed using the *MaAsLin2* default "LM" analysis method. Output from *MaAsLin2* was transformed to a log scale, and statistical significance was considered at the qValue ≤0.05 level. Visualization of statistically significant differential abundance was generated using the *ggplot2* package. When a large number of genera obtained a statistically significant differential abundance, a subset of key genera known to be associated with food safety or spoilage was reported. A full list of all genera that were significantly of greater abundance can be found in the supplementary data.

## RESULTS

### ASV contamination can alter the observed microbiome composition in low-biomass meat samples

After qPCR amplification, the mean 16S rRNA copy number was 504 copies/µL per meat sample (range 23–4,574), 65 copies/µL per negative control (range 2–222), 110 copies/µL per sterile water sample (range 98–123), and 117,201 copies/µL for the one positive control. Twenty-three percent of all meat samples had a copy number less than 100 copies/µL. These results suggest that raw meat samples should be considered low-biomass samples for purposes of microbiome investigations.

Across all 150 samples, 13.3 M raw paired-end reads were generated, including samples from both meat types, negative controls, one positive control, and sterile water blanks. The average number of raw reads was 91,808 reads per meat sample (range 12,820–181,566), 4,041 reads per negative control (range 2,750–3,517), 16,250 reads per sterile water sample (range 889–31,612), and 114,622 reads for the one positive control. A total of 6.68 M reads remained after quality control filtering and merging of forward and reverse reads in the *DADA2* package. The package *decontam* was then used to identify potential contaminant ASVs using three different methods: frequency, prevalence, and combined.

The frequency method is a best-fit determination of a predicted frequency-based contaminant and non-contaminant model to determine ASVs that are considered potential contaminants (17). The frequency method identified was run at an initial threshold = 0.1 and removed 124 ASVs as contaminants, leaving 11,554 for further analysis. The removed ASVs were minor contributors to overall abundance, with 0.32% of total sample reads removed as contaminants using the frequency method.

The prevalence method is based on the assumption that contaminants will likely be higher in the negative controls than in true samples, and a chi-squared statistic is calculated between true samples and the negative controls to identify potential contaminants (17). The prevalence method was conducted at an initial threshold of 0.5. At this threshold, it removed 155 ASVs total, leaving 11,523 for further analysis. In contrast to the frequency method, one of the ASVs removed by the prevalence method appeared to be a significant contributor to the overall 16S count abundance in a subset of samples. Across all samples, this ASV was the 32nd most abundant ASV based on the count matrix. Further investigation into this ASV showed that it was of the genus *Listeria* (Fig. S3), which could potentially be found in raw meat samples collected from a production plant facility, thus suggesting it may not be a contaminant. Upon further investigation, this ASV was detected in only 10 of the 150 samples, with 8 of those 10 samples being from meat, as well as one negative control, and the positive control from the mock community. Additionally, all but one of the meat samples and both the positive and negative control were part of the same DNA extraction batch, i.e., extraction batch 13. Importantly, this *Listeria* ASV was the most abundant component within the mock community and was also found in highest abundance in the positive control sample, followed by the adjacent samples extracted along with the positive control in extraction batch 13. As a result, it was determined that this contaminant ASV likely originated from the positive control and was introduced into the negative controls and the samples in this particular extraction batch during a low-level cross contamination event. Thus, it was determined that this ASV should indeed be removed as a likely contaminant.

The combined method combines the frequency- and prevalence-based scores into a composite score to determine potential contaminants (17). The initial combined method was run at a threshold of 0.40, resulting in the removal of 311 ASVs, leaving 11,367 for microbiome analysis (Table S1). A majority of the ASVs were not high-abundance contributors to the overall microbiome profile. However, the *Listeria* ASV that was assigned as a contaminant in the prevalence method was also determined to be a contaminant using the combined method. Further iterations of the combined method at lower thresholds also determined the *Listeria* ASV to be a contaminant. As a result, the combined method was used to remove contaminant ASVs from the final count matrix.

In total, 6.66 M paired-end sequence reads remained after removal of the likely contaminant ASVs. Additionally, no significant differences were observed in the average number of reads generated for the BJ and TL meat samples (Fig. S1).

## Characterization of identified taxa in the positive control and meat samples

The positive control generated 114,622 reads, and 66,512 remained after quality control analysis in *DADA2*. The positive control was not subjected to *decontam* to remove potential contaminants. Seven genus-level taxa were assigned to the sequences in the positive control data using the Zymo reference database (Fig. S2). Neither *Staphylococcus* nor *Enterococcus* was recovered in the mock community positive control, likely due to their low overall relative abundance in the mock control (0.0001% and 0.0007%, respectively). Additionally, *Pseudomonas* was overrepresented at 41.0% of the mock positive control (theoretical abundance = 2.8%), and *Listeria* was underrepresented at 50.8% of the mock positive control (theoretical abundance = 95.9%).

After removing negative controls, the positive control, and sterile water blanks, 6.52 M reads were classified into a total of 11,367 ASVs (Table S1). ASVs were classified to Bacteria (96.97%), Archaea (0.14%), and NA (2.89%) at the kingdom level. The percentage of reads able to be classified against the Silva 138.1 database decreased as the taxonomic

level decreased, resulting in 86.1% read classification rate at the genus level but only 13.41% at the species level. Classification rates for ASVs were even lower, with 53.8% of all ASVs identified at the genus level and only 6.40% identified at the species level. Based on these results, microbiome analysis was conducted at the genus level.

## Processing date and meat type are associated with microbiome variation

To determine microbial community differences between meat type and key production schedule factors, beta diversity was assessed using PERMANOVA and NMDS performed on Bray-Curtis distances. The variance in beta diversity partitioned to each measured study variable was statistically significant for all individual factors in the PERMANOVA model (Table 1). In terms of magnitude of effect, meat type ($R^2 = 0.06$, $P = 0.001$), day of the week ($R^2 = 0.06$, $P = 0.001$), and processing date ($R^2 = 0.14$, $P = 0.001$) were associated with the largest variance in microbiome beta diversity, representing 83.9% of the total explained variation by individual factors. Other processing schedule factors such as shift time ($R^2 = 0.02$, $P = 0.005$) and test period ($R^2 = 0.03$, $P = 0.001$) accounted for a smaller amount of observed variation in beta diversity. The interaction of meat type:shift was significantly associated with microbiome beta diversity ($R^2 = 0.02$, $P = 0.001$). Although the interaction of meat type:processing date also had an association with microbiome beta diversity ($R^2 = 0.14$), it was not determined to be statistically significant ($P = 0.161$). NMDS ordination of the Bray-Curtis distances was also performed (Fig. 1) to visualize the composition of the microbial communities by processing date.

To further understand the impact of processing date on meat microbiome diversity, a time series analysis was conducted across each processing date based on the estimated marginal means for aerobic plate count (APC) and alpha diversity measurements (Fig. S4). The BJ values tended to be higher than the TL values for APC, observed richness, Shannon's diversity, and evenness across all the processing dates. However, the magnitude of difference between the TL and BJ on each processing date was inconsistent due to variation in APC and alpha diversity on each date. Additionally, the daily trend for APC and alpha diversity was not parallel between the BJ and TL. Divergence in the trend line between the BJ and TL was evident on specific days and weeks, suggesting the diversity of the meat microbiome may be influenced by other factors beyond the inherent meat microbiome of the BJ and TL, and those factors may not be consistent between processing dates.

The abundance of taxa in each meat sample was also highly variable between individual meat samples at both the phylum and genus levels as measured by sequencing read counts (Fig. S5 and S6) and relative abundance (Fig. S7; Fig. 2). In general, Actinobacteria and Firmicutes had the highest number of reads at the phylum level, and *Pseudomonas* and *Acinetobacter* at the genus level. However, the genera identified as having either less than 100 reads or less than 1% relative abundance made up a large portion of the microbiome in both the BJ and TL, indicating a long tail of low-abundance bacteria within the meat microbiome.

**TABLE 1** Variance in beta diversity associated with key factors (*adonis2*)

| Factor | N | $R^2$ | P value |
|---|---|---|---|
| Day of the week | 4 | 0.06 | 0.001 |
| Test period (month) | 3 | 0.03 | 0.001 |
| Shift time | 3 | 0.02 | 0.005 |
| Meat type | 2 | 0.06 | 0.001 |
| Processing date | 24 | 0.14 | 0.001 |
| Meat:shift | | 0.02 | 0.01 |
| Meat:date | | 0.14 | 0.161 |
| Total $R^2$ | | 0.47 | |

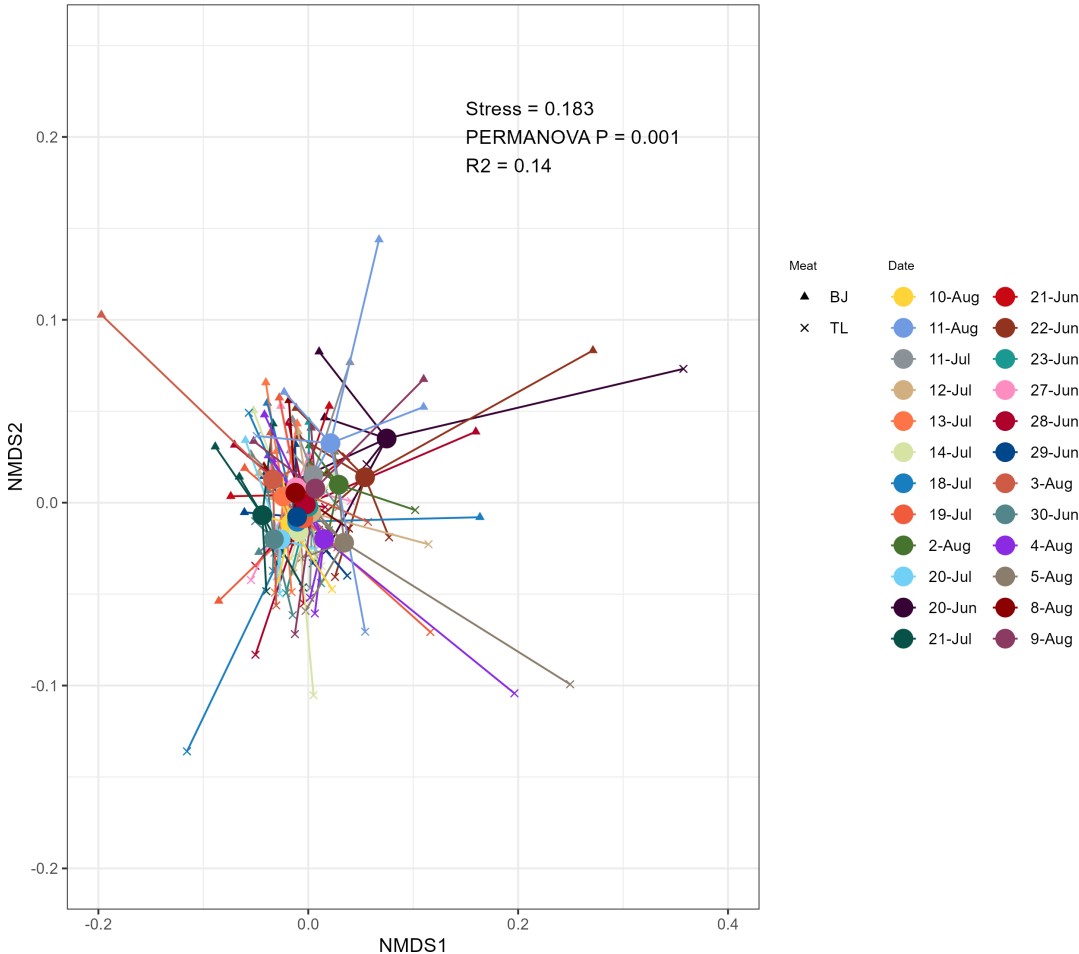

**FIG 1** Bray-Curtis ordination of beta diversity of both BJ and TL samples by processing date.

## Microbial ecology is different between meat types

Differences in the composition and diversity of the microbiome were observed between the BJ and TL. Visualization of beta diversity by NMDS ordination revealed significant clustering by meat type ($R^2 = 0.06$, $P = 0.001$), suggesting that the overall microbiome composition is dissimilar between the two meat types (Fig. 3A). Although there was no biologically significant difference in APC between the BJ and TL (3.12 vs 2.97, $P > 0.05$, Fig. S8A), the BJ samples had significantly higher observed richness (359 vs 290, $P < 0.001$, Fig. S8B), Shannon's diversity (4.44 vs 4.08, $P < 0.001$, Fig. S8C), and evenness (0.77 vs 0.73, $P = 0.003$, Fig. S8D) when compared to the TL.

The abundances of 89 genera were found to be significantly different between the BJ and TL, with 71 genera more abundant in the BJ and 18 genera more abundant in the TL (Data Set S1, "significant_results_DiffAb.csv"). This included several key bacteria associated with food safety and spoilage that were significantly more abundant in the BJ vs the TL at the qValue < 0.05 level (Fig. 3B). *Yersinia, Moraxella, Lactobacillus, Clostridium sensu stricto1, Campylobacter, and Bacteroides* had higher relative abundance in the BJ compared to the TL samples; while *Pseudomonas, Paracoccus, Micrococcus, Lactococcus, Enhydrobacter, and Bacillus* had higher relative abundance in the TL samples. This would suggest that compositional microbiome differences between the BJ and TL may be present at a very early time point in overall shelf life, and these differences could affect the food safety and quality risk profile of the two meat types.

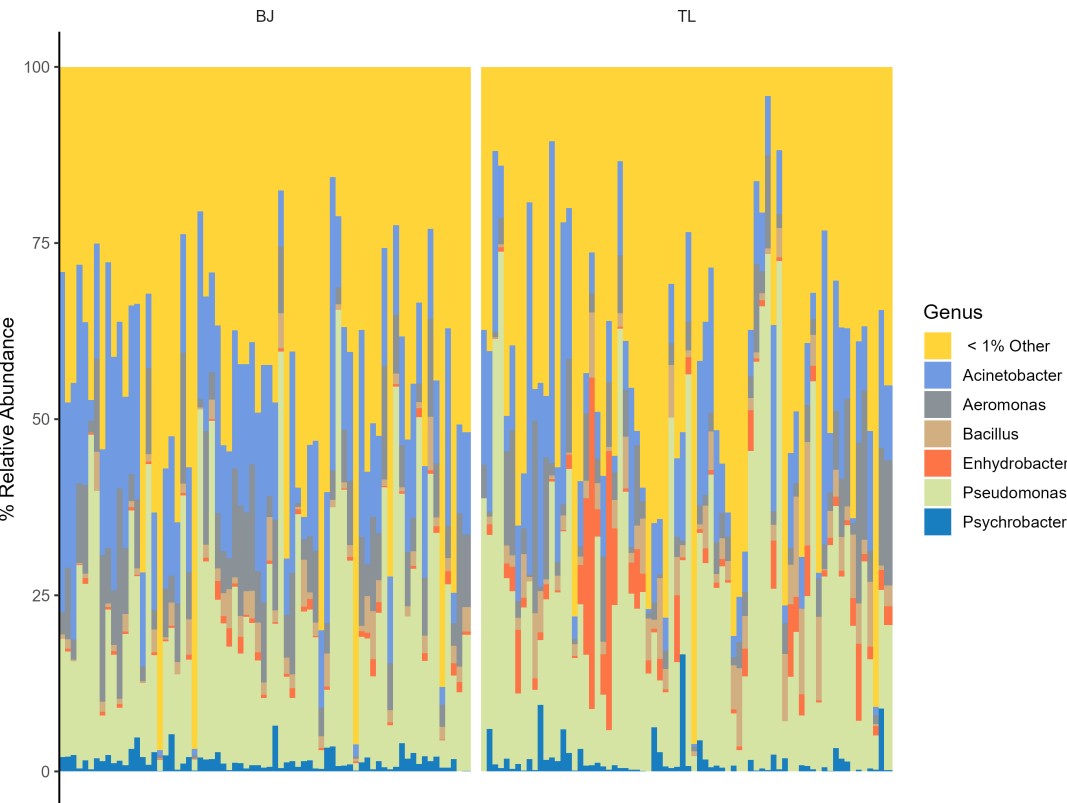

**FIG 2** Taxonomic relative abundance of individual meat samples by meat type at the genus level. Taxa with a median less than 1% relative abundance are cumulatively reported as "other." Each sample is represented by an individual bar on the *x*-axis. Individual sample information can be found in Figure S2.

## Changes in microbial ecology across a production shift are minimal but detectable

The composition of the meat microbiome remained relatively consistent throughout a processing shift, for both the BJ and TL samples. Overall, the interaction of meat type:shift was significantly associated with dissimilarity of the microbiome ($R^2 = 0.02$, $P = 0.001$). When samples were subsetted by individual meat type, the composition of the microbiome was also significantly associated with shift time in the BJ samples ($R^2 = 0.05$, $P = 0.001$) but was not associated with shift time in the TL samples ($R^2 = 0.03$, $P = 0.13$). NMDS ordination of beta diversity by shift time did not show clear clustering patterns in either the BJ or TL samples (Fig. 4A and C). A biologically significant difference was observed for APC in the TL throughout the shift, with the beginning of the shift being higher than the middle and end of the shift (3.38 log CFU/g vs 2.84 and 2.68, respectively, $P < 0.01$, Fig. 5A). While a similar decreasing trend was observed for APC in the BJ, differences were not statistically significant between the time points (3.31 log CFU/g vs 2.95 and 3.09, respectively, $P > 0.05$, Fig. 5A). An increasing trend was observed from the beginning to the middle and end of the shift in the BJ for observed richness (335 vs 356 and 386, respectively, Fig. 5B) and Shannon's diversity (4.35 vs 4.44 and 4.54, respectively, Fig. 5C), although none of these differences were statistically significant at the $P < 0.05$ level. The TL showed a similar increasing trend from the beginning to the middle of the shift for observed richness (259 vs 324, Fig. 5B) and Shannon's diversity (3.91 vs 4.23, Fig. 5C), but both observed (288) and Shannon's diversity (4.10) decreased at the end of the shift. However, as with the BJ samples, none of these differences were statistically significant at the $P < 0.05$ level. Evenness (Fig. 5D) remained essentially unchanged from the beginning, middle, and end of the shift for the BJ (0.77, 0.77, and 0.78, $P > 0.05$) and TL (0.73, 0.72, and 0.74, $P > 0.05$).

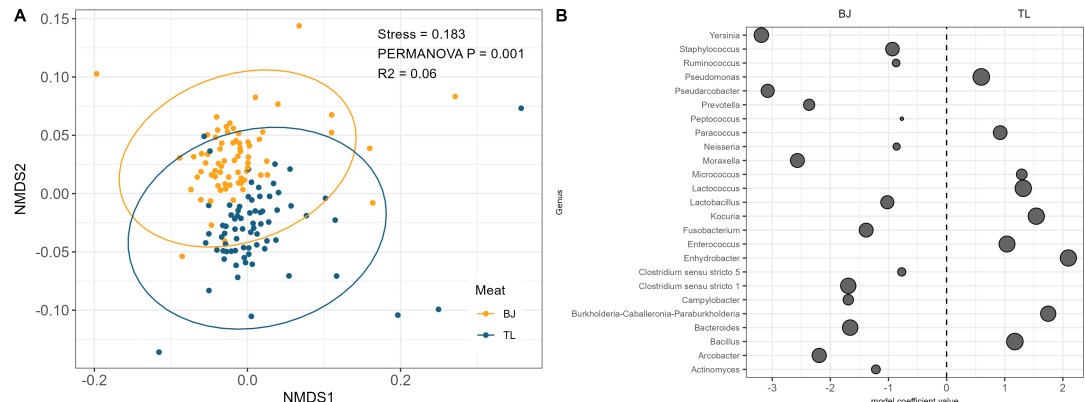

**FIG 3** Compositional description of the microbiome by meat type. (A) Non-metric multidimensional scaling plot of Bray-Curtis ordination of Bootjack Trim (yellow) and tenderloin (blue). (B) Differences in abundance of select microbial genera between Bootjack Trim (negative) and tenderloin (positive) that are contributors to food safety or shelf life. Each dot represents a microbial genus that is significantly different between the two meat types, with the size of the dot representing the overall prevalence of the genus.

The genus-level relative abundance profiles of meat samples at each shift time were highly variable at the individual sample level, similar to what was observed between the two meat types. For both the BJ and TL, the dominant genera in a majority of samples were *Pseudomonas*, *Acinetobacter, and Aeromonas,* and this dominance was consistent throughout the shift. Within the lower-abundance genera, *Streptococcus* and *Escherichia–Shigella* were significantly more abundant at the end of the shift than at the beginning of the shift in the BJ samples (Fig. 4B). Only *Empedobacter* was found to have a higher abundance at the beginning of the shift in the TL samples (Fig. 4D). None of the higher-abundance genera displayed statistically significant differences in abundance throughout the shift in either the BJ or TL samples.

## *Salmonella* prevalence and enumeration were low and centered around specific processing dates

Overall *Salmonella* prevalence was higher in the BJ compared to the TL (8.3% vs 4.2%, respectively, Table 2), but this difference was not statistically significant ($P$ = 0.48). Prevalence of *Salmonella* in the BJ was higher at the middle of the shift (12.5%) than at the beginning and end of the shift (4.2% and 8.3%, respectively), but the differences were not statistically significant ($P$ = 0.55). No statistically significant differences were observed across shift time in TL (4.2% for each time point, $P$ = 1.0). All three TL samples that tested positive for *Salmonella* by qPCR also showed the presence of low-abundance *Salmonella* ASVs through 16S analysis. In contrast, none of the six *Salmonella*-positive BJ samples detected by qPCR had *Salmonella* ASVs identified via 16S.

Enumeration of *Salmonella* was low in all *Salmonella*-positive meat samples, with 8 of 9 positive samples having an estimated enumeration of either 0–1 CFU/g or <1 CFU/g (Fig. 6A). The detection of *Salmonella* in both the BJ and TL appeared to cluster around two distinct processing windows (Fig. 6B). The first cluster was found on Julian dates 172 and 173 (21 June and 22 June) with three BJ samples testing positive for *Salmonella* on these two dates (one sample on date 172 and two samples on date 173). The second cluster was found on Julian dates 200 and 202 (19 July and 21 July). One positive sample was found in BJ on 19 July, followed by three positive samples in TL and two positive samples in BJ on 21 July. *Salmonella* was not detected in any other meat samples throughout the duration of this study.

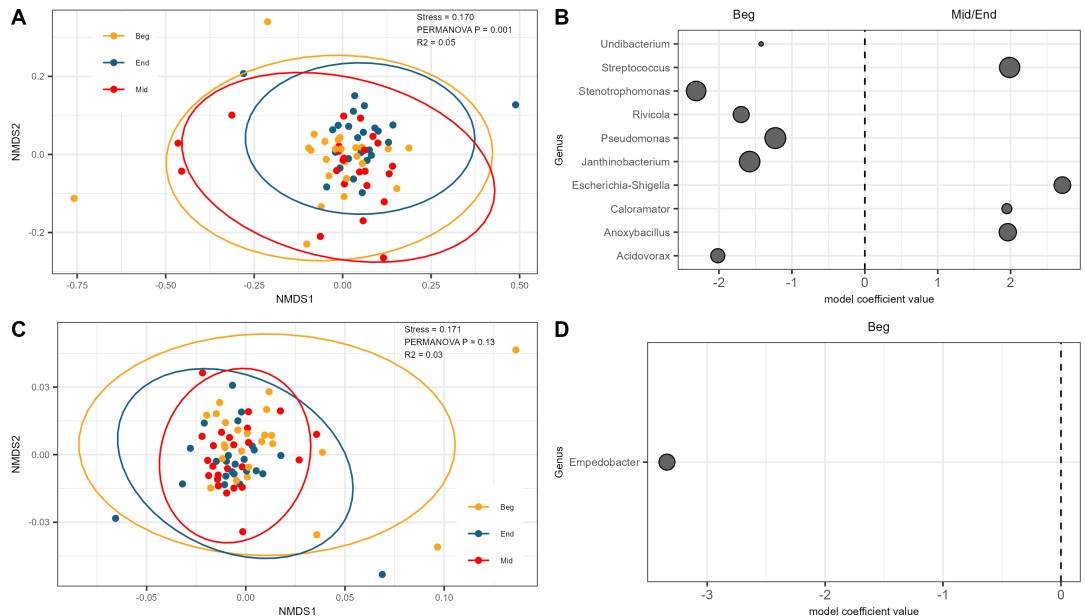

**FIG 4** Compositional description of the microbiome by meat type and shift time. (A) Non-metric multidimensional scaling plot of Bray-Curtis ordination of Bootjack Trim at the beginning (yellow), middle (red), and end (blue) of the production shift. (B) Difference in abundance of microbial genera between the beginning of shift (negative) and either middle or end of shift (positive) in the Bootjack Trim. (C) Non-metric multidimensional scaling plot of Bray-Curtis ordination of tenderloin at the beginning (yellow), middle (red), and end (blue) of the production shift. (D) Difference in abundance of microbial genera between the beginning of shift (negative) and either middle or end of shift (positive) in the tenderloin. Each dot represents a microbial genus that is significantly different between the two meat types, with the size of the dot representing the overall prevalence of the genus.

## DISCUSSION

### Fresh pork microbial ecology is variable by processing date

Microbiome analysis using 16S sequencing and culture-based aerobic plate count revealed a diverse microbiome with compositional differences associated with individual processing dates. In addition to microbiome variation associated with each hog being harvested, other potential daily sources of variation exist throughout the entire harvest process, which may affect the day-to-day carcass surface microbiome. Lairage and stunning (3, 7, 8, 20), various subsequent processing steps (5) and pork-specific harvest interventions (8, 20, 21), evisceration (22), and anatomical location on the carcass (22, 23) may all influence the daily variation of the fresh pork microbiome. Variability in daily sanitation practices, temperature, and handling by workers may also influence the degree to which the processing environment contributes to the microbial ecology of the meat surface (2, 21, 24, 25).

The importance of processing date also arose in the results related to *Salmonella* in both the BJ and TL samples, as *Salmonella* prevalence seemed to cluster around specific processing dates (Fig. 6B). Although both the overall prevalence and load of *Salmonella* in each of these raw materials were lower than what has been previously reported (9), this study provides evidence that the presence of *Salmonella* in fresh pork raw materials may not be homogeneously detected throughout a production schedule and that certain processing dates have the potential for increased prevalence of *Salmonella*. While outside the scope of the current study, variation in *Salmonella* prevalence by individual processing date may be related to the prevalence of *Salmonella* on farms that supply pigs on given dates (26), highly invasive *Salmonella* serotypes (27), shedding and cross contamination between pigs during transport and lairage (28, 29), processing line contamination during evisceration (22), carcass chilling and fabrication (30), and daily sanitation practices (31, 32).

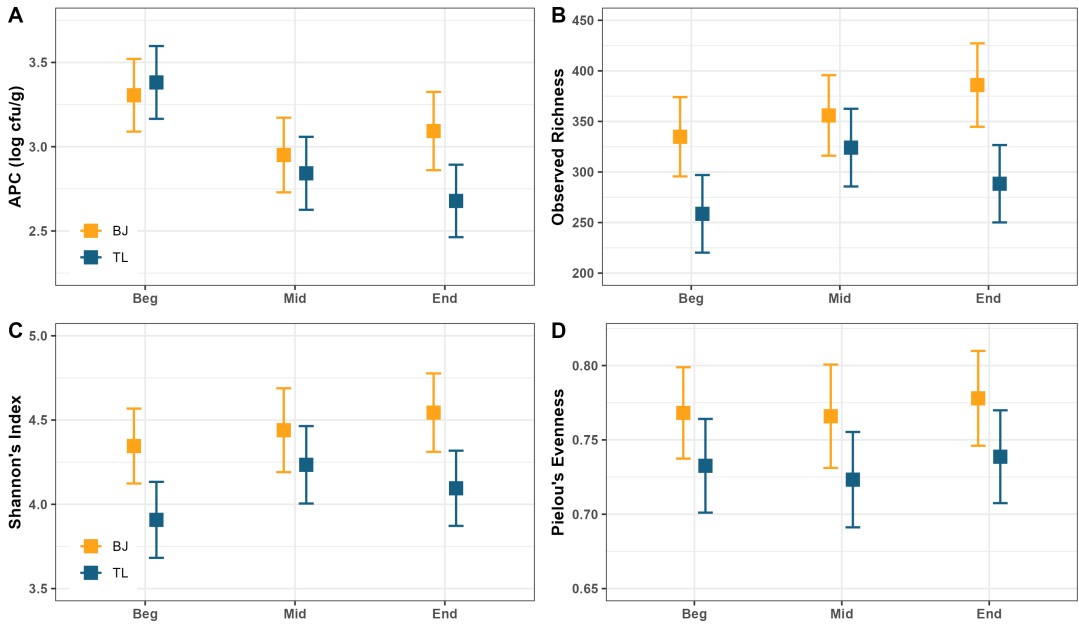

**FIG 5** Microbiome diversity of the Bootjack Trim ("BJ," yellow) and tenderloin ("TL," blue) across the production shift. Estimated marginal means and 95% confidence intervals for (A) aerobic plate count, (B) observed richness, (C) Shannon's diversity index, and (D) Pielou's evenness.

A limitation of this pilot study was that *Salmonella* isolates were not further investigated for serotype, sequence type, or SNP analysis from whole genome sequencing. This limited our ability to determine the relatedness of *Salmonella*-positive samples identified across the production schedule and to determine potential sources of *Salmonella* in the meat. Further research is needed to assess daily variation in both the live animals and harvest/fabrication process and determine their relation to the prevalence of *Salmonella* in fresh pork raw materials.

## Fresh pork microbial ecology is variable by meat type

A microbiome assessment using 16S sequencing revealed that both the BJ and TL contained highly diverse yet distinct microbial communities, which were also highly variable at an individual sample level. Interestingly, microbiome diversity was significantly different between the BJ and TL samples, while APC was not (Fig. 3; Fig. S8), suggesting that microbiome analysis can reveal bacterial population differences in raw meat samples that are not detected using culture-based methods such as APC. The more granular resolution of microbiome data and its subsequent capacity to differentiate samples has been previously reported. For example, a recent study characterized the

**TABLE 2** *Salmonella* prevalence of Bootjack Trim and tenderloin across shift times

| Meat | *Salmonella* | | |
| | N | Positive_samples | Prevalence |
| --- | --- | --- | --- |
| Bootjack Trim – shift time | | | |
| Beginning | 24 | 1 | 4.2% |
| Middle | 24 | 3 | 12.5% |
| End | 24 | 2 | 8.3% |
| Overall | 72 | 6 | 8.3% |
| Tenderloin – shift time | | | |
| Beginning | 24 | 1 | 4.2% |
| Middle | 24 | 1 | 4.2% |
| End | 24 | 1 | 4.2% |
| Overall | 72 | 3 | 4.2% |

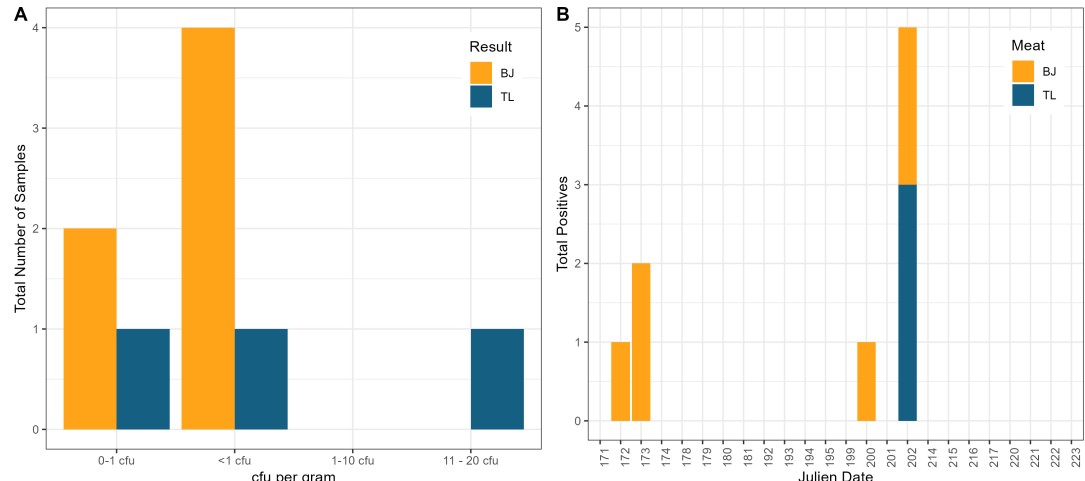

**FIG 6** Presence of *Salmonella* across a production schedule. (A) *Salmonella* enumeration of qPCR-positive Bootjack Trim (yellow) and tenderloin (blue) samples. (B) Histogram of total *Salmonella*-positive Bootjack Trim (yellow) and tenderloin (blue) samples by Julien sampling date.

environmental microbiome of multiple fabrication lines in a pork processing facility and observed significant differences in both alpha and beta diversity between the main production line, loin, picnic shoulder, belly, and ham fabrication lines (4). A study by Peruzy et al. determined that the microbiome of different anatomical areas of eight pork carcasses was typically dominated by the same genera, but differences in the micro-biome existed between slaughter processes and environmental exposure in different facilities (33). Additionally, Braley et al. found similar carcass surface microbiota were observed following evisceration in 26 individual carcasses that originated from different farms and sampled at multiple carcass locations (23). Taken together with the current study, these results may suggest that while microbial ecology may be similar across different anatomical locations on the carcass, each fabrication line may have its own distinct microbiome that adds to the microbiome of the meat surface as the carcass is fabricated into primal and subprimal cuts. Further studies are needed to understand the individual fabrication line environmental microbiome and its relationship with the associated meat microbiome.

In addition to differences in microbiome composition, the BJ tended to have a higher crude prevalence of *Salmonella* than the TL (Table 2), although it did not achieve statistical significance. This is in agreement with Scott et al. (9) that showed a lower prevalence of *Salmonella* in intact and non-intact pork cuts vs comminuted pork. Given the overall low number of *Salmonella* positives in this study, the lack of statistical significance could potentially be a result of Type II error. Further research is needed to assess potential differences in *Salmonella* prevalence between the BJ, TL, and other fresh pork cuts.

## Implications for food safety and quality

Improved understanding of both high- and low-abundance taxa in raw pork may provide insights for both the safety and quality of fresh pork. The predominant microflora observed in meat samples collected in this study was similar to what has been previ-ously reported on the carcass surface (4, 23, 33, 34). These predominant taxa are often known to be related to the spoilage of raw pork. However, a high degree of sample-to-sample variation in relative abundance profiles was observed in both the BJ and TL, both in high-abundance taxa and low-abundance taxa that individually accounted for <1% relative abundance in the sample (Fig. 2). Sample-to-sample variation observed in key spoilage bacteria such as *Pseudomonas, Aeromonas, Bacillus, Enhydrobacter, and Acinetobacter* should be a consideration for targeted interventions both upstream from

the fabrication line (i.e., evisceration and into the carcass cooler) and also for the formulation of effective antimicrobial systems in the finished product.

Differential abundance analysis revealed differences between the BJ and TL for several low-abundance taxa, including several that are considered to be key drivers of food safety and quality of fresh pork (Fig. 3B). While these taxa may be in low abundance at this initial point in the pork production chain, the potential for outgrowth and a shift in the overall relative abundance of these taxa exists throughout the shelf life of the product as the changing environment (e.g., cold chain, antimicrobials, frozen storage, thermal processing, packaged shelf life, etc.) may either select for or limit the growth of the genera within the microbiome. It is important to note that the microbiome represented in this study reflects a starting point for the evolution of the microbial ecology on the meat surface, prior to exposure to the rest of the supply cold chain, further processing, and packaged shelf life.

The compositional differences observed in this study are of importance given the final usage of these two meat types specifically. The TL is typically sold raw with a relatively short refrigerated shelf life that is often supplemented by the use of antimicrobials and/or packaging structures such as modified atmosphere packaging or vacuum packaging. The greater abundance of key spoilage bacteria such as *Pseudomonas, Paracoccus, Lactococcus,* and *Lactobacillus* in the TL (Fig. 3B) underscores the need to control spoilage bacteria in the TL through either fabrication line sanitation strategies, effective packaging systems, and/or through formulation with an effective antimicrobial system. While BJ is typically further processed into ground pork trim to be used in sausage formulations, the belly primal cut from which it is excised is processed into bacon using a non-ready-to-eat thermal process that relies on certain formulation and processing parameters to limit the outgrowth of either spoilage bacteria or pathogens during the process or over the shelf life of the product. The greater abundance of *Yersina, Clostriduium sensu strito 1, Campylobacter,* and *Staphylococcus* found in the BJ would suggest that a different food safety risk potential exists between the BJ and TL (Fig. 3B), and that cold chain control of the outgrowth of these pathogens during shelf life and further processing is critical as the BJ is processed into fresh sausage, dry sausage, or bacon from the belly primal cut. Additionally, the greater abundance of gut bacteria such as *Escherichia coli–Shigella* and *Streptococcus* at the middle/end of the shift in the BJ may suggest that factors associated with the fabrication line or process may influence the presence of these bacteria in the BJ meat that is not associated with a similar increase in the TL (Fig. 4B). As a result, targeted intervention strategies on the BJ fabrication line may have a greater impact on food safety and quality than on the TL fabrication line.

## Meat rinsate samples had low microbial biomass and were vulnerable to contamination

Contamination can be problematic in microbiome studies that analyze samples with low microbial biomass (18, 35, 36). Although the meat samples analyzed in this study had a mean APC of approximately 3.0 log CFU/g, the low copy numbers obtained via qPCR for the 16S gene suggest that the meat samples had relatively low numbers of bacterial and archaeal genomes. It is important to note that the APC values are obtained by culturing viable bacteria, whereas the 16S qPCR copy numbers are obtained from non-cultured, total extracted DNA; and thus they are not directly comparable. Sources of contamination can include the sample collection process, DNA extraction, and library preparation, including nucleic acids contained in extraction and library preparation reagents. Care must be taken at each of these steps to minimize the possibility of either environmental or sample-to-sample contamination. Because meat samples have low biomass, the use of negative controls to assess contamination is strongly recommended, and our results demonstrate the utility of such negative controls. Specifically, our use of negative controls and an algorithmic decontamination procedure revealed that a majority of meat samples and one negative control from a single DNA extraction batch were contaminated with an ASV that was highly abundant in the positive control that was run as part

of that same extraction batch. Such an analysis would not have been possible without negative controls and careful documentation of extract batches. In addition to negative controls and algorithmic identification of potential contaminants using *decontam*, it is important to integrate biological knowledge into contaminant removal. Preliminary understanding of the sample microbial ecology allows for a biologically based assessment of ASVs that would be expected in the particular sample type analyzed. This study identified a *Listeria* ASV as a contaminant that since all samples taken were from raw meat in a raw pork fabrication plant, it could not be automatically ruled out as a contaminant. Further analysis at the individual sample level revealed a logical route for this *Listeria* ASV to be introduced and was deemed appropriate to be identified as a contaminant. Thus, an understanding of the contaminant profile can be valuable in determining the overall impact of the ASVs removed and providing a more accurate representation of the microbiome presented for further statistical analysis.

## Conclusions

Taken in total, this study presents evidence that the microbial ecology is dissimilar between two different meat types collected from two different fabrication lines across a production schedule, within a single pork plant. However, there is a very high amount of sample-to-sample variability within meat type, and individual processing dates have a strong association with both the dissimilarity of the microbiome and the presence of *Salmonella* in the meat sample. While it is impossible for processors to control every potential source of microbial variation present in the harvest and fabrication process, this study provides evidence that processors have a vital need to limit points of variation in the process that may contribute to the microbiome variation. Controlling these factors may result in positive effects on both food safety and shelf life quality of the fresh pork produced.

## ACKNOWLEDGMENTS

This work was funded by Hormel Foods Corporation. The authors declare the following financial interests/personal relationships which may be considered as potential competing interests: A.E.A., S.M.S., and K.J.S. are employees of Hormel Foods Corporation.

## AUTHOR AFFILIATIONS

[1]Department of Veterinary Population Medicine, University of Minnesota, St. Paul, Minnesota, USA
[2]Hormel Foods Corporation, Austin, Minnesota, USA

## AUTHOR ORCIDs

A. E. Asmus  http://orcid.org/0000-0003-1585-9072
T. N. Gaire  http://orcid.org/0000-0003-1805-8051
N. R. Noyes  http://orcid.org/0000-0001-6149-1008

## AUTHOR CONTRIBUTIONS

A. E. Asmus, Conceptualization, Data curation, Formal analysis, Funding acquisition, Investigation, Methodology, Project administration, Resources, Software, Supervision, Validation, Visualization, Writing – original draft | T. N. Gaire, Formal analysis, Investigation, Methodology, Software, Writing – review and editing | K. J. Schweisthal, Conceptualization, Methodology, Writing – review and editing | S. M. Staben, Conceptualization, Methodology, Writing – review and editing | N. R. Noyes, Conceptualization, Formal analysis, Investigation, Methodology, Project administration, Resources, Supervision, Writing – review and editing

## DATA AVAILABILITY

All data sets are available at the NCBI SRA database under BioProject accession number PRJNA1156799.

## ADDITIONAL FILES

The following material is available online.

### Supplemental Material

**Data Set S1 (Spectrum02209-24-s0001.csv).** All differentially abundant genera between meat types, significant at q value < 0.05.

**Data Set S2 (Spectrum02209-24-s0002.csv).** Taxonomic relative abundance of individual meat samples by meat type at the genus level.

**Data Set S3 (Spectrum02209-24-s0003.csv).** Taxonomic abundance of sequencing reads of individual meat samples at the phylum level.

**Data Set S4 (Spectrum02209-24-s0004.csv).** Taxonomic abundance of sequencing reads of individual meat samples at the genus level.

**Data Set S5 (Spectrum02209-24-s0005.csv).** Taxonomic relative abundance of individual meat samples by meat type at the phylum level.

**Supplemental material (Spectrum02209-24-s0006.pdf).** Figure S1 to S8; Table S1.

### Open Peer Review

**PEER REVIEW HISTORY (review-history.pdf).** An accounting of the reviewer comments and feedback.

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
