## [Reviewer comments · Microbiology Spectrum]

Microbiology Spectrum

Microbiome Characterization of Two Fresh Pork Cuts During Production in a Pork Fabrication Facility

Aaron Asmus, Tara N Gaire, Kyle Schweisthal, Sally Staben, and Noelle Noyes

Corresponding Author(s): Noelle Noyes, University of Minnesota Twin Cities

Review Timeline:

Submission Date:	September 6, 2024
Editorial Decision:	October 15, 2024
Revision Received:	December 4, 2024
Accepted:	December 30, 2024

Editor: Salina Parveen

Reviewer(s): Disclosure of reviewer identity is with reference to reviewer comments included in decision letter(s). The following individuals involved in review of your submission have agreed to reveal their identity: Anuradha J. Punchihewage Don (Reviewer #1)

Transaction Report:

DOI: <https://doi.org/10.1128/spectrum.02209-24>

Re: Spectrum02209-24 (Microbiome Characterization of Two Fresh Pork Cuts During Production in a Pork Fabrication Facility)

Dear Dr. Noelle Noyes:

Thank you for the privilege of reviewing your work. Below you will find my comments, instructions from the Spectrum editorial office, and the reviewer comments.

Revision Guidelines

Sincerely,
Salina Parveen
Editor
Microbiology Spectrum

Reviewer #1 (Public repository details (Required)):

The BioProject is publicly available in the NCBI under accession number PRJNA1156799,

Reviewer #1 (Comments for the Author):

Microbiome Characterization of Two Fresh Pork Cuts During Production in a Pork Fabrication Facility

Salmonella and other microbial contaminants in fresh pork pose significant food safety challenges. This review examines the microbial profiles of Bootjack Trim (BJ) and Tenderloin (TL) cuts using 16S rRNA sequencing, focusing on microbial diversity and Salmonella prevalence within a commercial production environment. Overall, the manuscript is well-written. The Materials and Methods section is particularly well-explained, providing crucial detail for future researchers to accurately reproduce the data. The other sections are also well-explained and include the necessary information for thorough understanding. However, the authors should work on enhancing the figure quality, as some figures have low resolution.

Please find my comments below.

Page 5: 2nd line: You can delete the extra "that"

Page 7: In scientific writing, it's standard practice to write temperatures with no space between the degree symbol and the unit. Please revise '4.5{degree sign} C' to '4.5 {degree sign}C' for consistency in entire manuscript.

Page 7: Please ensure consistency in formatting units. '1 mL' and '9mL' should both follow the standard format with a space between the number and the unit, so revise '9mL' to '9 mL'.

Page 8: Can you clarify the reason for using 4.5 {degree sign}C instead of the more standard 4 {degree sign}C?

Page 8: Please consider revising the phrase 'Each bag was aggressively hand massaged' to something more precise and professional, such as 'Each bag was vigorously shaken by hand.'

Page 8: Please revise the phrase 'Limit of Detection of (LOD) of 1' to 'Limit of Detection (LOD) of 1' to avoid redundancy.

Page 17: line 8: *Listeria* should be italicized.

The figure quality in the manuscript is low. Please work on enhancing the resolution and clarity of the figures to meet publication standards.

Figure 1: Needs to be enhanced due to low resolution.

Figure 2: The X-axis labels are difficult to read due to overlapping, please adjust the spacing or formatting.

Figure 3-A also requires enhancement for clarity.

Figure 4-A and C need improvement.

Figure 6: Needs to be enhanced.

This applies to the supplemental materials as well.

Reviewer #2 (Public repository details (Required)):

The data set has been deposited and is publicly available.

Reviewer #2 (Comments for the Author):

- In this study, the authors investigated the microbial ecology of pork cuts in a new pork production plant. In particular, the authors sought to determine microbial profiles at different stages of pork production, Salmonella prevalence and for two different pork cuts over multiple timepoints in the production schedule.
- The authors stated that the findings would be useful for implementing "target intervention strategies" to improve food safety. Overall, this study seems nicely done with experimental methods with all considerations of the appropriate controls. However, much of the results section reads more like a methods paper. This reviewer felt bogged down in the controls minutiae and had trouble grasping any significance to the findings or whether this study might be useful to the pork processing industry. It was not clear to this reviewer how these findings will be used to "target intervention strategies."
- Additionally, it was hard to see a good flow of the story and how the microbiome analysis and Salmonella detection by qPCR are connected.
- Something that did seem significant in the findings was the "greater abundance of...Yersinia, Clostridium....etc. in the BJ meat" implying different food safety risks. Perhaps this is useful information for pork factories to consider for their decontamination/antimicrobial strategies.
- My impression of the impact of the processing dates on the microbiota and prevalence of Salmonella is that this was probably caused by the animals being processed, not necessarily factory practices, and the authors may want to mention this in the discussion in the text.
- The abstract looks a little long. The authors should check to make sure the abstract does not exceed the word limit. I think it would be more reader-friendly to be concise and stick to the main findings for the abstract.
- This reviewer felt the discussion was too long and hand wavy. It was a lot of work for the reviewer to find the main conclusion points. This section needs to be trimmed down and made more succinct. If the journal allows it, I would suggest breaking into smaller sections with headings.

Additional comments relating to figures:

- 1) Figure 1: I just see samples categorized by processing date. It might be more informative if you used different symbols for BJ vs. TL samples.
- 2) Sample names on the X axis of Figure 2 cannot be read due to the tiny font.
3. Also, it could be informative to have the authors discuss any significance about detecting a lot of *Pseudomonas*, *Enhydrobacter*, *Bacillus*, *Aeromonas*, and *Acinetobacter* in several of these samples. Are these also a concern to take into consideration? Should these higher abundance microbes also be considered during antimicrobial strategies employed at the

factory? It seems that some of these organisms actually detected in the microbiome analysis could at times have a higher impact on food safety than lower abundance, occasionally detected Salmonella.

4) Figure 3: If I understand this correctly, Bootjack Trim negative, is on the left side of the plot and Tenderloin positive is on the right side? If that is the case, I would also label the meat types on the left and right side of the plot in panel B. It is already very technical so it is nice to make it as ready friendly with labeling as possible so that people aren't guessing what you mean and are grasping the most important things you want them notice.

5) Figure 4: How does the compositional shift over time potentially inform factory operations from antimicrobial perspective? Is there a certain shift that should perform specialized cleaning? Also, detection of Escherichia and Shigella in the Bootjack Trim seems like it would also be a huge concern Figure 4, Panel B.

6) Figure 6 and Table 2. Bootjack Trim seems to have slightly higher prevalence than TL. Why is that and what are the implications for the factory.

7) Table 2., Salmonella should be in italics.

8) Supplemental S7. I do not see Salmonella genus listed. Did it fall into the other reads category?

Microbiome Characterization of Two Fresh Pork Cuts During Production in a Pork Fabrication Facility

Salmonella and other microbial contaminants in fresh pork pose significant food safety challenges. This review examines the microbial profiles of Bootjack Trim (BJ) and Tenderloin (TL) cuts using 16S rRNA sequencing, focusing on microbial diversity and *Salmonella* prevalence within a commercial production environment. Overall, the manuscript is well-written. The Materials and Methods section is particularly well-explained, providing crucial detail for future researchers to accurately reproduce the data. The other sections are also well-explained and include the necessary information for thorough understanding. However, the authors should work on enhancing the figure quality, as some figures have low resolution.

Please find my comments below.

Page 5: 2nd line: You can delete the extra “that”

Page 7: In scientific writing, it’s standard practice to write temperatures with no space between the degree symbol and the unit. Please revise ‘4.5° C’ to ‘4.5 °C’ for consistency in entire manuscript.

Page 7: Please ensure consistency in formatting units. ‘1 mL’ and ‘9mL’ should both follow the standard format with a space between the number and the unit, so revise ‘9mL’ to ‘9 mL’.

Page 8: Can you clarify the reason for using 4.5 °C instead of the more standard 4 °C?

Page 8: Please consider revising the phrase ‘Each bag was aggressively hand massaged’ to something more precise and professional, such as ‘Each bag was vigorously shaken by hand.’

Page 8: Please revise the phrase ‘Limit of Detection of (LOD) of 1’ to ‘Limit of Detection (LOD) of 1’ to avoid redundancy.

Page 17: line 8: *Listeria* should be italicized.

The figure quality in the manuscript is low. Please work on enhancing the resolution and clarity of the figures to meet publication standards.

Figure 1: Needs to be enhanced due to low resolution.

Figure 2: The X-axis labels are difficult to read due to overlapping, please adjust the spacing or formatting.

Figure 3-A also requires enhancement for clarity.

Figure 4-A and C need improvement.

Figure 6: Needs to be enhanced.

This applies to the supplemental materials as well.

- In this study, the authors investigated the microbial ecology of pork cuts in a new pork production plant. In particular, the authors sought to determine microbial profiles at different stages of pork production, *Salmonella* prevalence and for two different pork cuts over multiple timepoints in the production schedule.
- The authors stated that the findings would be useful for implementing “target intervention strategies” to improve food safety. Overall, this study seems nicely done with experimental methods with all considerations of the appropriate controls. However, much of the results section reads more like a methods paper. This reviewer felt bogged down in the controls minutiae and had trouble grasping any significance to the findings or whether this study might be useful to the pork processing industry. It was not clear to this reviewer how these findings will be used to “target intervention strategies.”
- Additionally, it was hard to see a good flow of the story and how the microbiome analysis and *Salmonella* detection by qPCR are connected.
- Something that did seem significant in the findings was the “greater abundance of...*Yersinia*, *Clostridium*....etc. in the BJ meat” implying different food safety risks. Perhaps this is useful information for pork factories to consider for their decontamination/antimicrobial strategies.
- My impression of the impact of the processing dates on the microbiota and prevalence of *Salmonella* is that this was probably caused by the animals being processed, not necessarily factory practices, and the authors may want to mention this in the discussion in the text.
- The abstract looks a little long. The authors should check to make sure the abstract does not exceed the word limit. I think it would be more reader-friendly to be concise and stick to the main findings for the abstract.
- This reviewer felt the discussion was too long and hand wavy. It was a lot of work for the reviewer to find the main conclusion points. This section needs to be trimmed down and made more succinct. If the journal allows it, I would suggest breaking into smaller sections with headings.

Additional comments relating to figures:

- 1) Figure 1: I just see samples categorized by processing date. It might be more informative if you used different symbols for BJ vs. TL samples.
- 2) Sample names on the X axis of Figure 2 cannot be read due to the tiny font.
3. Also, it could be informative to have the authors discuss any significance about detecting a lot of *Pseudomonas*, *Enhydrobacter*, *Bacillus*, *Aeromonas*, and

Acinetobacter in several of these samples. Are these also a concern to take into consideration? Should these higher abundance microbes also be considered during antimicrobial strategies employed at the factory? It seems that some of these organisms actually detected in the microbiome analysis could at times have a higher impact on food safety than lower abundance, occasionally detected *Salmonella*.

4) Figure 3: If I understand this correctly, Bootjack Trim negative, is on the left side of the plot and Tenderloin positive is on the right side? If that is the case, I would also label the meat types on the left and right side of the plot in panel B. It is already very technical so it is nice to make it as ready friendly with labeling as possible so that people aren't guessing what you mean and are grasping the most important things you want them notice.

5) Figure 4: How does the compositional shift over time potentially inform factory operations from antimicrobial perspective? Is there a certain shift that should perform specialized cleaning? Also, detection of *Escherichia* and *Shigella* in the Bootjack Trim seems like it would also be a huge concern Figure 4, Panel B.

6) Figure 6 and Table 2. Bootjack Trim seems to have slightly higher prevalence than TL. Why is that and what are the implications for the factory.

7) Table 2., *Salmonella* should be in italics.

8) Supplemental S7. I do not see *Salmonella* genus listed. Did it fall into the other reads category?

Reviewer #1 (Comments for the Author):

Microbiome Characterization of Two Fresh Pork Cuts During Production in a Pork Fabrication Facility

Salmonella and other microbial contaminants in fresh pork pose significant food safety challenges. This review examines the microbial profiles of Bootjack Trim (BJ) and Tenderloin (TL) cuts using 16S rRNA sequencing, focusing on microbial diversity and Salmonella prevalence within a commercial production environment. Overall, the manuscript is well-written. The Materials and Methods section is particularly well-explained, providing crucial detail for future researchers to accurately reproduce the data. The other sections are also well-explained and include the necessary information for thorough understanding.

Response: Thank you for your helpful review, it has substantially improved the manuscript.

However, the authors should work on enhancing the figure quality, as some figures have low resolution.

Response: Thank you for comment in regards to the lower resolution on figures in the main manuscript and supplemental material. Figures in the initial submission were copied from .png files into a .doc file. All figures have now been saved individually as high resolution .tiff files according to the requirements of Microbiology Spectrum.

Please find my comments below.

Page 5: 2nd line: You can delete the extra "that"

Response: Thank you for noticing the grammatical error on Page 5. The extra word has been deleted and the change highlighted.

Page 7: In scientific writing, it's standard practice to write temperatures with no space between the degree symbol and the unit. Please revise '4.5{degree sign} C' to '4.5 {degree sign}C' for consistency in entire manuscript.

Response: Thank you for noticing the stylistic error regarding °C. The 15 instances where this occurred have been corrected.

Page 7: Please ensure consistency in formatting units. '1 mL' and '9mL' should both follow the standard format with a space between the number and the unit, so revise '9mL' to '9 mL'.

Response: Thank you for highlighting the format error on Page 7. The change to the correct format for 9 mL has been made and is highlighted.

Page 8: Can you clarify the reason for using 4.5 {degree sign}C instead of the more standard 4 {degree sign}C?

Response: Thank you for noticing the temperature of the holding refrigerators. It is part of the standard operating procedures at the Hormel Foods Research and Development Laboratory for walk-in coolers and refrigerators to operate at 40 °F (4.5 °C).

Page 8: Please consider revising the phrase 'Each bag was aggressively hand massaged' to something more precise and professional, such as 'Each bag was vigorously shaken by hand.'

Response: Thank you for the feedback regarding the phrasing of this sentence. The suggested phrasing of this sentence has been included and is highlighted.

Page 8: Please revise the phrase 'Limit of Detection of (LOD) of 1' to 'Limit of Detection (LOD) of 1' to avoid redundancy.

Response: Thank you for highlighting the repeated word in this sentence. The repeated word has been removed from this sentence and is highlighted.

Page 17: line 8: *Listeria* should be italicized.

Response: Thank you for noticing the stylistic error in this sentence. The change to italicize *Listeria* has been made and is highlighted.

The figure quality in the manuscript is low. Please work on enhancing the resolution and clarity of the figures to meet publication standards.

Figure 1: Needs to be enhanced due to low resolution.

Response: Thank you for highlighting the need to improve this Figure 1. Per journal instructions this figure has been uploaded as a high resolution .tiff file at 300 dpi. Additionally, the BJ and TL samples have been designated by different symbols (BJ = "x", TL = "▲") to provide further clarity to the figure.

Figure 2: The X-axis labels are difficult to read due to overlapping, please adjust the spacing or formatting.

Response: Thank you for highlighting the need to improve the clarity of Figure 2. The X-axis Sample ID labels have been removed from the figure, further description has been given to the legend, and a supplemental file has been added that provides % relative abundance for each individual sample that coincides with each bar on the figure.

Figure 3-A also requires enhancement for clarity.

Response: Thank you for highlighting the need to improve this Figure 3. Per journal instructions this figure has been uploaded as a high resolution .tiff file at 300 dpi. Figure 3B has

also been edited to improve the clarity of the X - axis. Additionally, labels have been added at the top of Figure 3B to provide further clarity which genera are of higher abundance in each meat type.

Figure 4-A and C need improvement.

Response: Thank you for highlighting the need to improve this Figure 4. Per journal instructions this figure has been uploaded as a high resolution .tiff file at 300 dpi. Figures 4A and 4C have been improved by increasing the resolution and reducing the font size of the legend, as well as the X and Y axis labels, to enhance the overall clarity of the figures.

Figure 6: Needs to be enhanced.

Response: Thank you for highlighting the need to improve this Figure 6. Per journal instructions this figure has been uploaded as a high resolution .tiff file at 300 dpi.

This applies to the supplemental materials as well.

Response: Thank you for highlighting the need to improve the supplemental figures. Any changes that were made in Figures 1- 5 that would also provide clarity to the supplemental figures were also made.

Reviewer #2 (Comments for the Author):

- In this study, the authors investigated the microbial ecology of pork cuts in a new pork production plant. In particular, the authors sought to determine microbial profiles at different stages of pork production, Salmonella prevalence and for two different pork cuts over multiple timepoints in the production schedule.

Response: Thank you for your helpful review, it has substantially improved the manuscript.

- The authors stated that the findings would be useful for implementing "target intervention strategies" to improve food safety. Overall, this study seems nicely done with experimental methods with all considerations of the appropriate controls. However, much of the results section reads more like a methods paper. This reviewer felt bogged down in the controls minutiae and had trouble grasping any significance to the findings or whether this study might be useful to the pork processing industry. It was not clear to this reviewer how these findings will be used to "target intervention strategies."

Response: Thank you for providing this valuable insight. We also agree that the importance of targeted intervention strategies deserves more discussion, which we now have included in **Lines 580 - 585 and Lines 614 - 620.**

- Additionally, it was hard to see a good flow of the story and how the microbiome analysis and Salmonella detection by qPCR are connected.

Response: Thank you for your feedback regarding the connection between the microbiome and the detection of *Salmonella*. While this study wasn't statistically powered to establish a potential link between the fresh pork microbiome and the presence of *Salmonella*, it does reveal differences in the microbiome between meat types and shows that these differences are

temporally dynamic throughout the processing schedule. We've added content to this point in **Lines 580 - 585 and Lines 614 - 620**.

- Something that did seem significant in the findings was the "greater abundance of...Yersinia, Clostridium....etc. in the BJ meat" implying different food safety risks. Perhaps this is useful information for pork factories to consider for their decontamination/antimicrobial strategies.

Response: Thank you for noticing this potentially important finding. We agree it deserves some discussion, which we now have included in **Lines 580 - 58 and Lines 614 - 620** .

- My impression of the impact of the processing dates on the microbiota and prevalence of Salmonella is that this was probably caused by the animals being processed, not necessarily factory practices, and the authors may want to mention this in the discussion in the text.

Response: Thank you for providing this valuable insight. We agree it deserves some discussion, which we now have included in **Lines 511 - 512 and Lines 537 - 539**.

- The abstract looks a little long. The authors should check to make sure the abstract does not exceed the word limit. I think it would be more reader-friendly to be concise and stick to the main findings for the abstract.

Response: Thank you for prompting a word count evaluation of the Abstract and Introduction. While the Abstract is in line with the requirements of the journal (213/250 words), the Importance section exceeded the word count (154/150 words). Edits to reflect the change in the Importance section have been made and are highlighted.

- This reviewer felt the discussion was too long and hand wavy. It was a lot of work for the reviewer to find the main conclusion points. This section needs to be trimmed down and made more succinct. If the journal allows it, I would suggest breaking into smaller sections with headings.

Response: Thank you for prompting us to be more concise. We agree with this feedback and changes have been made in **Lines 514 - 519, 527 - 532, and other areas highlighted throughout the Discussion**. Additionally, the Discussion section has been split from 2 sections to 5 sections to help readers easily identify the main conclusion points and implications. Several paragraphs were moved or removed to coincide with the heading changes and improve conciseness.

Additional comments relating to figures:

1) Figure 1: I just see samples categorized by processing date. It might be more informative if you used different symbols for BJ vs. TL samples.

Response: Thank you for highlighting the need to improve Figure 1. The BJ and TL samples have been designated by different symbols (BJ = "x", TL = "▲") to provide further clarity to the figure.

2) Sample names on the X axis of Figure 2 cannot be read due to the tiny font.

Response: Thank you for highlighting the need to improve the clarity of Figure 2. The X-axis Sample ID labels have been removed from the figure, further description has been given to the

legend, and a supplemental file has been added that provides % relative abundance for each individual sample that coincides with each bar on the figure.

3. Also, it could be informative to have the authors discuss any significance about detecting a lot of *Pseudomonas*, *Enhydrobacter*, *Bacillus*, *Aeromonas*, and *Acinetobacter* in several of these samples. Are these also a concern to take into consideration? Should these higher abundance microbes also be considered during antimicrobial strategies employed at the factory? It seems that some of these organisms actually detected in the microbiome analysis could at times have a higher impact on food safety than lower abundance, occasionally detected *Salmonella*.

Response: Thank you for noticing this potentially important finding. We agree it deserves some discussion, which we now have included in **Lines 580 - 585**.

4) Figure 3: If I understand this correctly, Bootjack Trim negative, is on the left side of the plot and Tenderloin positive is on the right side? If that is the case, I would also label the meat types on the left and right side of the plot in panel B. It is already very technical so it is nice to make it as ready friendly with labeling as possible so that people aren't guessing what you mean and are grasping the most important things you want them notice.

Response: Thank you for highlighting the need to improve Figure 3. Figure 3B has also been edited to improve the clarity of the X - axis. Additionally, labels have been added at the top of Figure 3B to provide further clarity to which genera are of higher abundance in each meat type.

5) Figure 4: How does the compositional shift over time potentially inform factory operations from antimicrobial perspective? Is there a certain shift that should perform specialized cleaning? Also, detection of *Escherichia* and *Shigella* in the Bootjack Trim seems like it would also be a huge concern Figure 4, Panel B.

Response: Thank you for these questions, which are important to address. We address these in new **Lines 614 - 620**.

6) Figure 6 and Table 2. Bootjack Trim seems to have slightly higher prevalence than TL. Why is that and what are the implications for the factory.

Response: We agree this is an important consideration, and we have added a discussion in **Lines 564 - 571**.

7) Table 2., *Salmonella* should be in italics.

Response: Thank you for noticing the stylistic error in this sentence. The change to italicize *Salmonella* has been made and is highlighted.

8) Supplemental S7. I do not see *Salmonella* genus listed. Did it fall into the other reads category?

Response: Thank you for noticing that *Salmonella* was not included in Supplemental S7. *Salmonella* was identified by 16S in 3 individual TL samples that coincided with the results of the BAX/Hygiene qPCR positive tests. However, these samples had a low abundance of reads, falling below the 100-read threshold after median normalization of the read counts.

Salmonella was not detected by 16S in any other samples. This finding has been included in the Results section, **Lines 493 - 496**.

Re: Spectrum02209-24R1 (Microbiome Characterization of Two Fresh Pork Cuts During Production in a Pork Fabrication Facility)

Dear Dr. Noelle Noyes:

Your manuscript has been accepted, and I am forwarding it to the ASM production staff for publication. Your paper will first be checked to make sure all elements meet the technical requirements. ASM staff will contact you if anything needs to be revised before copyediting and production can begin. Otherwise, you will be notified when your proofs are ready to be viewed.

Sincerely,
Salina Parveen
Editor
Microbiology Spectrum

Reviewer #1 (Comments for the Author):

The revised manuscript has unnecessary spaces in the beginning of the sentences. I mentioned a few of them, however the authors need to go through the entire manuscript and fix them.
See below line numbers

Line numbers: 18,19,21,22, 25, 33, 35, 37

Reviewer #2 (Comments for the Author):

The revision looks great to this reviewer. I did notice on line 453, there are two "When" at the beginning of the sentence that should be fixed.